# RECURSIVE DEEP INVERSE REINFORCEMENT LEARNING

## ABSTRACT

Inferring an adversary's goals from exhibited behavior is crucial for counterplanning and non-cooperative multi-agent systems in domains like cybersecurity, military, and strategy games. Deep Inverse Reinforcement Learning (IRL) methods based on maximum entropy principles show promise in recovering adversaries' goals but are typically offline, require large batch sizes with gradient descent, and rely on first-order updates, limiting their applicability in real-time scenarios. We propose an online Recursive Deep Inverse Reinforcement Learning (RDIRL) approach to recover the cost function governing the adversary actions and goals. Specifically, we minimize an upper bound on the standard Guided Cost Learning (GCL) objective using sequential second-order Newton updates, akin to the Extended Kalman Filter (EKF), leading to a fast (in terms of convergence) learning algorithm. We demonstrate that RDIRL is able to recover cost and reward functions of expert agents in standard and adversarial benchmark tasks. Experiments on benchmark tasks show that our proposed approach outperforms several leading IRL algorithms.

## 1 INTRODUCTION

Inverse Optimal Control (IOC) and Inverse Reinforcement Learning (IRL) aim to infer parameterized cost and reward functions in optimal control and reinforcement learning problems, respectively, from observed state-control data. This data is assumed to be generated by an expert following an optimal policy that either minimizes a cost function or maximizes a reward function.

Previous IRL approaches have included maximum-margin approaches (Abbeel & Ng, 2004), and probabilistic approaches such as (Ziebart et al., 2008). In this work, we build on the maximum entropy IRL framework presented previously (Ziebart et al., 2008). In this framework, training consists of two nested loops. The inner loop approximates the optimal control policy for a hypothesized cost function, while the outer loop minimizes a negative log-likelihood cost function (Ziebart et al., 2008), constructed by sampling a full trajectory from the inner loop's optimal control policy and by using the expert trajectory that is observed from the expert.

Due to this nested structure, training under the maximum entropy deep IRL in an online fashion becomes very challenging since inner and outer loops need long trajectories and large batch sizes to converge. Available IRL approaches exploit the fact that it is often feasible to store and process entire state and control sequences in batches(Molloy et al., 2018). In real-time settings with memory, latency and compute constraints, this is generally not feasible.

Recursive optimization strategies such as Extended Kalman Filter (EKF) sequentially minimize a loss function that is a summation of mean square error of observed and estimated states, and mean squared error of the estimated states and their predicted values produced by assumed model dynamics (Humpherys et al., 2012). Hence, EKF cannot be naively leveraged to optimize the negative log-likelihood function (Ziebart et al., 2008) since the log of summation term could not be optimized sequentially. Recent works have proposed moment-matching approaches (Swamy et al., 2021; Zeng et al., 2022; 2025), leading to objective functions that have a simple summation form, making them more suitable for online adaptive learning. However, they are not explicitly derived from maximum entropy IRL, and prior formulations have not been optimized in an online setting.

To overcome this limitation, we require a reformulation of the maximum entropy objective into a structure amenable to recursive optimization. To address this gap, we show that the moment matching loss function introduced in (Swamy et al., 2021) provides an upper bound for the negative log-likelihood objective of maximum entropy IRL (Finn et al., 2016b; Ziebart et al., 2008). We then propose a recursive optimization algorithm that minimizes the moment matching loss using expert demonstrations and sampled trajectories from the inner optimal control policy. This approach alleviates the need to optimize the negative log-likelihood cost function only after collecting all trajectories from the inner-loop policy and the expert. Instead, it enables incremental optimization, processing each expert observation as it arrives.

The main contribution of this work is a deep maximum entropy online IRL algorithm, Recursive Deep Inverse Reinforcement Learning (RDIRL), that learns nonlinear cost and reward functions parameterized by neural networks directly from expert demonstrations as they arrive. Unlike previous deep learning approaches, our method updates the inner control policy after each new expert sample, enabling online adaptation of policies. By processing state–action pairs sequentially, without storing or batching entire trajectories, RDIRL is well-suited for real-time applications with memory and latency constraints. Moreover, because the policy and cost updates occur incrementally, our approach converges significantly faster than competing IRL methods. We validate our approach in simulated benchmark tasks, demonstrating that it outperforms leading IRL methods.

## 2  RELATED WORK

IRL, also known as IOC(Finn et al., 2016b), aims to learn reward or cost functions from expert agents operating under optimal control or reinforcement learning policies. Several IOC methods have been developed to recover finite-horizon optimal control cost functions, including approaches based on Karush-Kuhn-Tucker (KKT) conditions(Zhang et al., 2019b;a; Puydupin-Jamin et al., 2012), Pontryagin's minimum principle (Molloy et al., 2022; 2020; Jin et al., 2020), and the Hamilton-Jacobi-Bellman equation (Pauwels et al., 2014; Hatz et al., 2012).

These methods typically follow a two-stage process: first, a feedback gain matrix is computed from state and control sequences using system identification techniques, and second, linear matrix inequalities are solved to recover the objective-function parameters from the feedback gain matrix. Online variations of IOC methods based on the Hamilton-Jacobi-Bellman equation (Zhao & Molloy, 2024; Molloy et al., 2018; 2020; Self et al., 2020c;a;b) have also been developed. However, both offline and online versions of these methods are generally limited to simple parameter estimation, assume partial knowledge of the expert's cost function, and do not incorporate deep neural network (Deep Neural Network (DNN)) representations of cost functions.

IRL approaches have also been proposed based on maximum margin (Abbeel & Ng, 2004; Ratliff et al., 2006) and maximum entropy (Ziebart et al., 2008; Boularias et al., 2011). Among these, maximum entropy IRL, as introduced by (Ziebart et al., 2008), has become one of the leading approaches. In this framework, optimization seeks to find reward or cost function parameters that maximize the likelihood of the observed expert trajectory under a maximum entropy distribution. This involves estimating a partition function from samples drawn from a background distribution that represents a control policy (Finn et al., 2016a; Fu et al., 2017), which is dependent on a parameterized cost function. The control policy may range from reinforcement learning (Ho & Ermon, 2016; Fu et al., 2017) to receding horizon optimal control (Xu et al., 2022).

Building on maximum entropy IRL, feature-based methods (Hadfield-Menell et al., 2016; Wu et al., 2020) model the reward function as an inner product between a feature vector $f$ and a parameter vector $\theta$. These methods have been successfully implemented, with the feature characteristics and parameter vector size typically chosen to match the true cost function structure. However, they assume some structural knowledge of the expert's cost function or domain knowledge (Finn et al., 2016b). Online versions of feature-based maximum entropy IRL have also been developed (Rhinehart & Kitani, 2018; Arora et al., 2021), but they have not yet been extended to include a DNN parameterization of the reward and cost functions.

Similarly, maximum entropy IRL with deep learning representations of the reward function has been successfully implemented (Wulfmeier et al., 2015). These methods, which leverage DNNs for complex reward functions, have gained popularity and become widely used (Finn et al., 2016b;

Wulfmeier et al., 2015; Ho & Ermon, 2016; Xu et al., 2019; 2022; Fu et al., 2017; 2019; Yu et al., 2019). As a result, they have emerged as leading IRL approaches, outperforming feature-based methods (Finn et al., 2016b; Xu et al., 2022; Ho & Ermon, 2016).

In this work, we propose a new online IRL method based on the maximum entropy framework (Ziebart et al., 2008; Ziebart, 2010). Unlike other online approaches (Molloy et al., 2018; Self et al., 2020c;b; Molloy et al., 2020; Rhinehart & Kitani, 2018; Arora et al., 2021), the proposed methodology allows the cost and reward functions to be parameterized using deep neural networks. Our approach is mostly related to the algorithm introduced by (Finn et al., 2016a), which minimizes a negative log-likelihood function and uses Model Predictive Path Integral Control (MPPI) (Xu et al., 2022) as the inner control policy. However, unlike prior work, we recursively adapt the sampling distribution representing the inner control policy each time an expert demonstration is observed.

To summarize, our proposed method is the first to combine several key features into a single effective algorithm. It can learn adversarial cost functions online, which is critical for applications such as evasion and pursuit. Additionally, it can learn complex, expressive cost functions, parameterized by deep neural networks, eliminating the need for manual design of cost-functions typically required in recursive methods (Molloy et al., 2018; Zhao & Molloy, 2024; Self et al., 2020c). While some prior methods have demonstrated good performance with online IOC(Zhao & Molloy, 2024; Molloy et al., 2020; Self et al., 2020c) and deep neural network-based cost functions(Finn et al., 2016a; Fu et al., 2017; Ho & Ermon, 2016; Zeng et al., 2022; Swamy et al., 2021), to the best of our knowledge, no previous approach has successfully combined these two properties.

## 3 BACKGROUND

### 3.1 MAXIMUM ENTROPY INVERSE REINFORCEMENT LEARNING

Our Inverse reinforcement learning method builds on Guided Cost learning framework Finn et al. (2016b) which is derived from maximum entropy Inverse Reinforcement Learning (IRL) (Ziebart et al., 2008). Our method seeks to learn an expert cost function or rewards function by observing the expert's behavior. The framework assumes the demonstrated expert behavior to be the result of the expert acting stochastically and near-optimally with respect to an unknown cost function. Specifically, the model assumes that the expert samples the demonstrated trajectories $\tau_i$ from the distribution (Finn et al., 2016b):

$$p(\tau) = \frac{1}{\mathcal{Z}} \exp(-c_\theta(\tau)) \tag{1}$$

where $\tau = \{x_1, u_1, \ldots, x_N, u_N\}$ is a trajectory sample, $x_N$ and $u_N$ are the agent's observed state and control input at time $N$ and $c_\theta(\tau) = \sum_{k=1}^N c_\theta(x_k, u_k)$ is an unknown cost function, parameterized by $\theta$, and associated with that trajectory.

The partition function $\mathcal{Z}$ is difficult to compute for large or continuous domains, and presents the main computational challenge in maximum entropy IRL. In the sample-based approach to maximum entropy IRL (Finn et al., 2016b; Fu et al., 2017; Ho & Ermon, 2016; Finn et al., 2016a) the partition function $\mathcal{Z} = \int \exp(-c_\theta(\tau))d\tau$ is estimated from a background distribution $q(\tau)$ representing the inner control policy, where $\tau$ are sampled from the policy $q(\tau)$. The central idea behind the maximum entropy approach is to estimate $\theta$ that maximizes the likelihood of the entropy cost distribution $p(\tau)$:

$$\hat{\theta} = \arg\max_\theta \quad p(\tau).$$

This approach is equivalent to minimizing the negative log-likelihood of Equation (1) given below (Finn et al., 2016b):

$$\mathcal{L}_{IRL}(\theta) = \frac{1}{N} \sum_{\tau_i \in \mathcal{D}_{\text{demo}}} c_\theta(\tau_i) + \log \frac{1}{M} \sum_{\tau_j \in \mathcal{D}_{\text{samp}}} \frac{\exp(-c_\theta(\tau_j))}{q(\tau_j)} \tag{2}$$

where $\mathcal{D}_{\text{samp}}$ is the set of $M$ background samples sampled from the inner control policy $q(\tau)$, $\mathcal{D}_{\text{demo}}$ is the set of $N$ expert demonstrations.

To represent the cost function $c_\theta(\tau)$, IOC or IRL feature-based methods typically use a linear combination of hand-crafted features $f : (u, x) \mapsto f(u, x)$, leading to $c_\theta(\tau) = \theta^T f(u_t, x_t)$ (Abbeel

& Ng, 2004). This representation is difficult to apply to more complex domains (Finn et al., 2016b). Recent works have focused on the use of high-dimensional expressive function approximators, representing $c_\theta(\tau)$ using neural networks, and outperforming feature-based methods (Finn et al., 2016b; Fu et al., 2017; Ho & Ermon, 2016). In this work, we only leverage neural networks to represent the cost function although, other parameterizations could also be used with our method. In practice, the negative log-likelihood in equation 2 is minimized using gradient descent and batch training. Previous algorithms using deep networks as the cost function parameterization required long and multiple expert demonstrations and sampled trajectories from background policies in order to converge through multiple training iterations. Moreover, training could not proceed before generating all expert and sampled trajectories which restricted it to offline training paradigms. In this work, we introduce a recursive optimization algorithm that adapts network parameters $\theta$ on the fly whenever an expert demonstration is observed.

### 3.2 Kalman Filtering

The Kalman Filter (KF) is among the most widely used state estimators in engineering applications. This algorithm recursively estimates the state variables, for example, the position and velocity of a projectile in a noisy linear dynamical system (Lipton et al., 1998), by minimizing the mean-squared estimation error of the current state, as noisy measurements are received and as the system evolves in time (Humpherys et al., 2012). Each update provides the latest unbiased estimate of the system variables. Since the updating process is fairly general and relatively easy to compute, the KF can often be implemented in real-time. When dealing with nonlinear systems extensions of the KF exist such as the EKF which resorts to linearizations using first-order Taylor's expansions Särkkä & Svensson (2023).

One interesting aspect is that the EKF can be seen as sequential second-order optimizer of cost functions of the form (Humpherys et al., 2012):

$$J_n(X_n, Y_n) = \sum_{k=1}^{n} j_k(x_k, y_k) \tag{3}$$

where $X_n = \{x_1, \ldots, x_n\}$ and $x_n$ represents the state of interest at time $n$. Moreover, $Y_n = \{y_1, \ldots, y_n\}$ where $y_n$ represents the measurement data at time $n$. $j_k$ represents the cost at time $k$ associated with $x_k$ and $y_k$, while $J_n$ is the cumulative value of $j_k$ and represents the cumulative cost associated with trajectories $X_n$ and $Y_n$. The EKF estimates the state $x_n$ that minimizes equation 3 at time $n$ using second-order Newton method as new measurement $y_n$ arrives. Thus, equation 3 can be re-written as:

$$J_n(X_n, Y_n) = J_{n-1}(X_{n-1}, Y_{n-1}) + j_n(x_n, y_n) \tag{4}$$

The EKF finds $x_n$ that minimizes equation 4 given previous loss function $J_{n-1}$, previous state estimates of $X_{n-1}$, previous measurements $Y_{n-1}$ and current measurement $y_n$. In classical Kalman filtering applications such as navigation and target tracking (Ward et al., 2006; Roumeliotis & Bekey, 2000), the goal is to estimate states $x_n$ given sequences of noisy (often Gaussian) data $y_n$. In this work, however, we aim at estimating the parameters $\theta$ of the cost function $c_\theta(\tau)$ from expert demonstration $\tau \in D_{\text{demo}}$ recursively. Inspired by the Kalman filter's sequential optimization approach described in (Humpherys et al., 2012), we develop a sequential optimization approach to find $\theta$ that maximizes the entropy $p(\tau)$.

## 4 Moment Matching as Upper Bound of the Negative Log-likelihood

In this section, we derive an upper-bound of the negative log-likelihood, leading to an optimization problem that is suitable for KF-like online estimation of the parameter vector $\theta$. That is, the resulting upper bound can be written following the same summation structure of equations 3 and 4. The log-sum term in equation 2 prevents direct recursive minimization, but the derived upper bound resolves this issue and enables sequential optimization.

In (Matkovic & Pecaric, 2007), the authors present a general variant of Jensen's inequality for convex functions as follows. Let $[a, b]$ be an interval in $\mathbb{R}$, $y_1, \ldots, y_N \in [a, b]$, and $p_1, \ldots, p_N$ be positive

real numbers such that $\sum_{n=1}^{N} p_n = 1$. If $f : [a, b] \to \mathbb{R}$ is convex on $[a, b]$, then:

$$\sum_{n=1}^{N} p_n f(y_n) - f\left(\sum_{n=1}^{N} p_n y_n\right) \leq \quad f(a) + f(b) - 2f\left(\frac{a+b}{2}\right) \tag{5}$$

Replacing the function $f$ by the negative log function, $f = -\log$ which is a convex function, equation 5 can be re-written as Matkovic & Pecaric (2007) :

$$\log\left(\sum_{n=1}^{N} p_n y_n\right) \leq \sum_{n=1}^{N} p_n \log(y_n) - \log(a) - \log(b) + 2\log\left(\frac{a+b}{2}\right) \tag{6}$$

In what follows, we will consider $N = M$ in equation 2 for the sake of compactness. Let's define $p_n$ and $y_n$ as follows :

$$p_n = \frac{1}{N} \quad \text{and} \quad y_n = \frac{\exp(-c_\theta(\tau_i^{\text{samp}}))}{q(\tau_i^{\text{samp}})} \tag{7}$$

where $\tau_{\text{samp}}$ is a trajectory sampled from $D_{\text{samp}}$ and let $y_n$ be defined over an interval $[a, b] \in \mathbb{R}$. By replacing equation 7 in equation 6 we get the following inequality:

$$\log \frac{1}{N} \sum_{i=1}^{N} \frac{\exp(-c_\theta(\tau_i^{\text{samp}}))}{q(\tau_i^{\text{samp}})} \leq \frac{1}{N} \sum_{i=1}^{N} (-c_\theta(\tau_i^{\text{samp}}) - \log q(\tau_i^{\text{samp}})) - K \tag{8}$$

where $K = log(a) + log(b) - 2log\left(\frac{a+b}{2}\right)$. Replacing equation 8 in equation 2 we can derive an upper bound of equation 2 as follows:

$$\mathcal{L}_{IRL}(\theta) = \frac{1}{N} \sum_{i=1}^{N} c_\theta(\tau_i^{\text{demo}}) + \log \frac{1}{N} \sum_{i=1}^{N} \frac{\exp(-c_\theta(\tau_i^{\text{samp}}))}{q(\tau_i^{\text{samp}})}$$

$$\leq \frac{1}{N} \sum_{i=1}^{N} c_\theta(\tau_i^{\text{demo}}) + \frac{1}{N} \sum_{i=1}^{N} (-c_\theta(\tau_i^{\text{samp}}) - \log q(\tau_i^{\text{samp}})) - K \tag{9}$$

$$\leq \frac{1}{N} \sum_{i=1}^{N} \left[ c_\theta(\tau_i^{\text{demo}}) - c_\theta(\tau_i^{\text{samp}}) - C \right]$$

where $C = \log q(\tau_i^{\text{samp}}) + K$ and $\tau_{\text{demo}}$ is a trajectory sampled from $D_{\text{demo}}$ representing expert's trajectory. Since $C$ and $N$ are independent from model parameters $\theta$, minimizing the upper bound of equation 2 is now equivalent to minimizing the following loss:

$$\mathcal{L}_{\text{UB}-\text{MM}} = \sum_{i=1}^{N} \left[ c_\theta(\tau_i^{\text{demo}}) - c_\theta(\tau_i^{\text{samp}}) \right]. \tag{10}$$

This upper bound has a particularly important consequence: it transforms the maximum entropy IRL objective into a moment-matching loss. This structure is equivalent to recent moment matching formulations in IRL (Swamy et al., 2021; Zeng et al., 2022; 2025), which replace the log-partition function of MaxEnt IRL with expectation-matching objectives between expert and policy distributions. Our derivation shows that moment matching losses, particularly the formulation in (Swamy et al., 2021), can be interpreted as an upper bound of the maximum entropy negative log-likelihood.

## 5 RECURSIVE DEEP INVERSE REINFORCEMENT LEARNING

In the previous section, we derived the upper bound of the negative log-likelihood cost described in equation 2 and showed it's equivalent to moment matching (Swamy et al., 2021). In this section, we seek to minimize the moment matching loss of equation 10 recursively. To do so, we re-write the EKF optimization problem using the loss function derived in equation 10 and a regularization term. Given an expert trajectory $\mathcal{D}_{\text{demo}} \triangleq \{\tau^{(0)}, \dots, \tau^{(N-1)}\}$ we seek to determine an optimal solution $\theta^*(t_i)$ starting from initial condition $\theta(t_0)$ by solving the following mathematical optimization function:

$$\mathcal{L}_N(\Theta_N) = \mathcal{L}_{UB-MM} + \frac{1}{2} \sum_{i=1}^{N} \|\theta(t_i) - \theta(t_{i-1})\|_{Q_\theta^{-1}}^2$$

$$= \sum_{i=1}^{N} \left[ c_\theta(\tau_i^{\text{demo}}) - c_\theta(\tau_i^{\text{samp}}) \right] + \frac{1}{2} \sum_{i=1}^{N} \|\theta(t_i) - \theta(t_{i-1})\|_{Q_\theta^{-1}}^2. \tag{11}$$

where the second term in the right-hand side of equation 11 is a regularization term typical to Bayesian filtering algorithms Imbiriba et al. (2022); Ghanem et al. (2025). In a similar fashion to Kalman filtering optimization process described in (Humpherys et al., 2012; Ghanem et al., 2023), we seek to determine optimal solution $\Theta_N^* = \{\theta^*(t_0), \dots, \theta^*(t_N)\}$ using the second-order Newton method sequentially, which recursively finds $\Theta_N^*$ given $\Theta_{N-1}^*$. Noticing that problem equation 11 can be broken into predictive and update problems, we can derive its recursive solution, which is detailed in Section B.4 of the Appendix, and leads to the result in Theorem 5.1.

**Theorem 5.1.** *Given $\hat{\theta}(t_{i-1}) \in \hat{\Theta}_{i-1}$ and known $P_{\theta_{i-1}} \in R^{d_\theta \times d_\theta}$ , the recursive equations for computing $\hat{\theta}(t_i)$ that minimizes (15) are given by the following:*

$$\hat{\theta}(t_i) = \hat{\theta}(t_i|t_{i-1}) - P_{\theta_i}\left(C_{\tau_{\mathrm{demo}}}(t_i) - C_{\tau_{\mathrm{samp}}}(t_i)\right) \tag{12}$$

*$P_{\theta_i}$ being the lower right block of $\left(\nabla^2 \mathcal{L}_i(\hat{\Theta}_{i|i-1})\right)^{-1}$ recursively calculated as :*

$$P_{\theta_i} = \left[(P_{\theta_{i-1}} + Q_\theta)^{-1} + \left(C_{\tau_{\mathrm{demo}}}^2(t_i) - C_{\tau_{\mathrm{samp}}}^2(t_i)\right)\right]^{-1} \tag{13}$$

*Proof.* using Lemma B.3 in (Humpherys et al., 2012), the lower block $P_{\theta_i}$ of $\left(\nabla^2 \mathcal{L}_i(\hat{\Theta}_{i|i-1})\right)^{-1}$ is calculated as in equation 13 □

As a consequence of Theorem (5.1), $\hat{\theta}(t_i)$ is computed according to equation 12 using $\hat{\theta}(t_{i-1})$. The entire training procedure is detailed in Algorithm 1, and a detailed description of Algorithm 1 is described in Section B.3 of the Appendix.

---

**Algorithm 1** Recursive Deep Inverse Reinforcement Learning

---

1: Initialize Cost function $c_\theta$ with parameters $\theta_{t_0}$
    **while** episodes < K **do**
2: Initialize inner policy $q(\tau)$
3: Initialize $P_{\theta_0}$ and $Q_\theta$
4: **for** $i = 1, 2, \dots, N$ **do**
5:    Observe one expert sample $\tau_i^{demo}$
6:    Sample one observation $\tau_i^{samp}$ from $q(\tau)$
7:    Evaluate the gradients $C_{\tau_{\mathrm{demo}}}(t_i)$ and $C_{\tau_{\mathrm{samp}}}(t_i)$
8:    Evaluate the hessians $C_{\tau_{\mathrm{demo}}}^2(t_i)$ and $C_{\tau_{\mathrm{samp}}}^2(t_i)$
9:    $\hat{\theta}(t_i) \leftarrow \hat{\theta}(t_{i-1}) - P_{\theta_i}\left(C_{\tau_{\mathrm{demo}}}(t_i) - C_{\tau_{\mathrm{samp}}}(t_i)\right)$
10:   $P_{\theta_i} \leftarrow \left[(P_{\theta_{i-1}} + Q_\theta)^{-1} + C_{\tau_{\mathrm{demo}}}^2(t_i) - C_{\tau_{\mathrm{samp}}}^2(t_i)\right]^{-1}$
11:   update $q(\tau)$ with respect to $c_\theta$ using any policy optimization method
12: **end for**
    episodes $\leftarrow$ episodes $+ 1$

---

## 6 EXPERIMENTS

We evaluate the proposed RDIRL algorithm in continuous control benchmarks from OpenAI Gym (Brockman, 2016) and MuJoCo (Todorov et al., 2012), as well as in an adversarial cognitive radar scenario (Potter et al., 2024; Haykin, 2006). We compare its performance against state-of-the-art inverse reinforcement learning and imitation learning methods, including Generative Adversarial Imitation Learning (GAIL) (Ho & Ermon, 2016), Guided Cost Learning (GCL) (Finn et al., 2016b), Adversarial Inverse Reinforcement Learning (AIRL) (Fu et al., 2017), SQIL (Reddy et al., 2020), and Maximum Likelihood Inverse Reinforcement Learning (ML-IRL) (Zeng et al., 2022), a moment-matching variant of IRL. Experiments are conducted in two regimes: batch mode (section 6), where

competing methods are trained in their standard setting with full trajectory batches, and streaming mode, where updates occur sample by sample (Appendix B.5).

Unlike reinforcement learning methods such as SAC(Haarnoja et al., 2018) or PPO(Schulman et al., 2017), which require large trajectory batches to converge and thus fail in streaming or real-time settings, our approach leverages MPPI (Williams et al., 2016) as the inner control policy. Since MPPI updates its actions at every time step, it is naturally suited for online IRL. In preliminary experiments, MPPI also provided stable performance and fast convergence unlike traditional RL policies when integrated into the RDIRL framework. Furthermore, preliminary experiments showed that competing IRL methods paired with their original RL inner policies failed to converge in streaming mode too. For consistency and fairness, we therefore adapt all competing methods to use MPPI as the inner policy in both batch and streaming comparisons.

Our results show that RDIRL consistently outperforms all benchmarked methods in recovering reward functions. Policies trained with rewards learned by RDIRL achieve optimal or near-optimal behavior significantly faster than competing approaches. Crucially, unlike existing methods which require large batches of expert trajectories and environment rollouts to converge, RDIRL leverages online adaptation. This enables efficient learning from streaming demonstrations, making it particularly well-suited for adversarial and time-limited scenarios.

Table 1: Comparison of normalized averaged reward values across all episodes for different Gym environments and methods.

| Methods | CartPole | MountainCar | HalfCheetah-v4 | Hopper |
|---|---|---|---|---|
| SQIL (Reddy et al., 2020) | $0.947 \pm 0.088$ | $-0.001 \pm 4.79$ | $-1.56 \pm 0.89$ | $0.799 \pm 0.15$ |
| GAIL (Ho & Ermon, 2016) | $0.934 \pm 0.058$ | $0.236 \pm 0.203$ | $-0.521 \pm 1.15$ | $0.714 \pm 0.08$ |
| GCL (Finn et al., 2016b) | $0.92 \pm 0.09$ | $0.247 \pm 0.19$ | $-0.226 \pm 1.27$ | $0.69 \pm 0.075$ |
| AIRL (Fu et al., 2017) | $0.953 \pm 0.069$ | $0.233 \pm 0.204$ | $-0.54 \pm 1.11$ | $0.709 \pm 0.084$ |
| ML-IRL(Zeng et al., 2022) | $0.938 \pm 0.093$ | $0.253 \pm 0.19$ | $-0.32 \pm 1.12$ | $0.648 \pm 0.06$ |
| RDIRL (ours) | $\mathbf{0.993 \pm 0.013}$ | $\mathbf{0.68 \pm 0.32}$ | $\mathbf{0.496 \pm 0.59}$ | $\mathbf{0.803 \pm 0.11}$ |

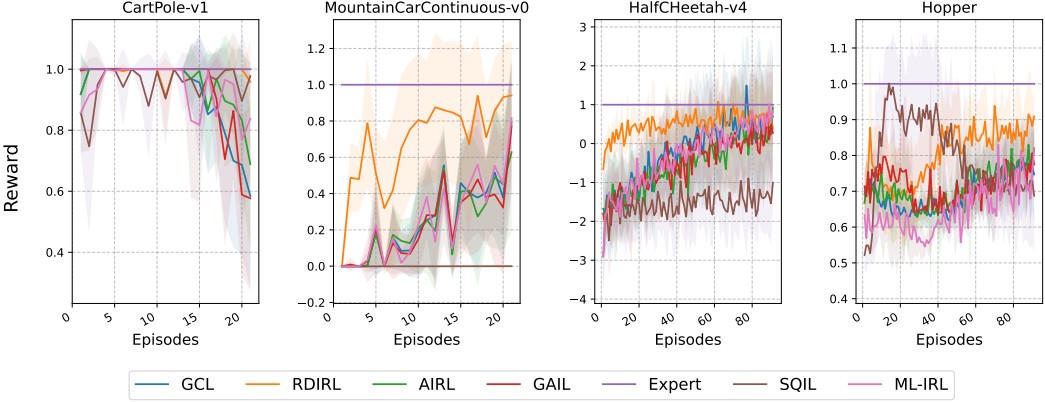

Figure 1: Learning curves for RDIRL and other methods.

## 6.1 CONTINUOUS CONTROL

To assess the performance of our proposed approach RDIRL, we conduct inverse reinforcement learning (IRL) experiments on the CartPole and Mountain Car environments from OpenAI Gym (Brock-

man, 2016) and HalfCheetah-v4, Hopper, and Walker2d from MuJoCo(Todorov et al., 2012), all solved using model-free reinforcement learning. Each task has a predefined true reward function provided by OpenAI Gym.

We first generate expert demonstrations for these tasks by training a PPO reinforcement learning agent (Schulman et al., 2017) to maximize the true reward function. Each expert demonstration consists of a state trajectory of size $N$ steps specified in Table 3 in B.1 for each task, which is then used as the sole expert trajectory for each IRL algorithm. Note that we do not use expert control sequences trajectory since we do not have access to the expert's control policy.

Next, we execute RDIRL to learn the reward function and train competing IRL algorithms using the expert trajectory over multiple episodes in batch mode, where each episode consists of an expert trajectory. This process is repeated for 12 Monte Carlo runs with different seeds. In all experiments, we use MPPI as the internal control policy $q(\tau)$ to maximize the learned reward function, $-c_\theta$. A detailed experiment description and parameter values of MPPI and IRL algorithms is described in Appendix B.1

We plot the mean of the normalized cumulative reward values across all episodes of trajectories $\tau^{\text{samp}}$ sampled from the inner control policy $q(\tau)$ in Figure 1.the averaged reward values are normalized with respect to the expert reward. In the case of RDIRL, $\tau^{\text{samp}}$ used to calculate the reward function in Figure 1 are generated online during training according to Algorithm 1. For the rest of the methods, $\tau^{\text{samp}}$ are generated offline after each offline training episode is completed.

All methods use the same neural network architecture to parameterize the reward function. Networks are randomly initialized at the start of each experiment, and all experiments are run on Nvidia-H200 GPU Cluster with 1 GPU per job(seed).

Our proposed method, RDIRL, successfully learns reward functions across all benchmark environments and consistently outperforms competing methods. In CartPole and MountainCar, it quickly recovers the expert reward even converging in one episode in CartPole, while in HalfCheetah and Hopper it achieves faster convergence and higher reward quality than baselines, many of which require far more episodes to converge or fail to converge. Learning curves in Figures 1 and 2 illustrate these improvements, with Walker2d results consistent with Figure 3 in (Reddy et al., 2020), where rewards closer to the expert indicate better performance. Furthermore, experimental results in streaming settings with detailed descriptions are provided in Appendix B.5.

Table 1 further shows that RDIRL achieves the highest normalized rewards in most tasks. This consistent outperformance stems from its recursive structure and adaptive uncertainty-aware updates, which improve sample efficiency and stability. Unlike traditional IRL, our method requires no fixed learning rate, as $P_\theta$ is updated at each step and acts as an adaptive rate.

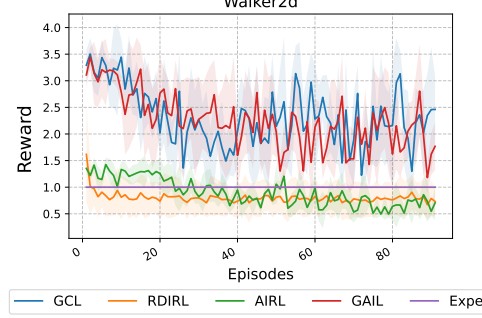

Figure 2: Learning curves for for Walker2d.

## 6.2 COGNITIVE RADAR

To evaluate whether our method can learn cost functions of adversarial agents, we perform inverse reinforcement learning experiments on a cognitive radar task. The task involves a radar chasing a moving target in 3D space. The target kinematic model follows constant velocity motion (Baisa, 2020) and the radar follows a second order unicycle model (Potter et al., 2024), where the target is moving linearly in space while the radar maximizes its Fisher Information Matrix (FIM) (Potter et al., 2024) to keep track of the target. Both the radar and the target live in the same 3D $x, y, z$ Cartesian plane. The goal of the target is to learn the radar's FIM from what it can observe from radar's states, which is in our case radar's position in 3D $x, y, z$ Cartesian coordinates. To achieve this goal using RDIRL,we execute Algorithm 1 where the radar's cost function is learned online. The radar's (expert) policy inside Algorithm 1 is an MPPI that maximizes radar's FIM. The inner control policy $q(\tau)$ is an MPPI that maximizes

the learned reward function, $-c_\theta$. The environmnent and IRL method's parameters are described in Table equation 3 inside the Appendix.

Furthermore, we compare RDIRL against GAIL, AIRL and GCL. To implement these methods, we generate expert trajectories for multiple episodes, where the expert policy is an MPPI that maximizes the radar's FIM. The inner control policy $q(\tau)$ in all of these baselines is an MPPI, with parameters specified in table 3. We repeat this process for 5 Monte Carlo runs using different seeds.

To test if the target successfully learned the radar's reward function, we plot the cumulative true FIM values resulting

Table 2: Comparison of mean FIM reward values for the Cognitive Radar example obtained by the different IRL methods.

| Methods | Mean Cumulative Reward |
|---|---|
| GAIL (Ho & Ermon, 2016) | 153.05 |
| GCL (Finn et al., 2016b) | 423.49 |
| AIRL (Fu et al., 2017) | 196.53 |
| RDIRL(ours) | **924.78** |

from the trajectories $\tau^{samp}$ sampled from the inner control policy $q(\tau)$ in Figure 3. We compare RDIRL's performance in learning the radar's reward function against GAIL, GCL, and AIRL. In the case of RDIRL, $\tau^{samp}$ used to calculate the reward function in Figure 1 are generated online during training according to algorithm 1. For the rest of the methods, $\tau^{samp}$ are generated offline after each offline training episode is completed.

In all algorithms, we used the same neural network architecture to parameterize the radar's FIM reward function: one hidden layer of 128 units, with a RELU activation function All networks were always initialized randomly at the start of each experiment and all experiments are run on on an intel core i7 CPU.

Results in Figure 3 show that RDIRL successfully learns the radar's FIM with a much faster convergence rate than the benchmark methods. The mean cumulative reward values across all episodes for each method are summarized in Table 2. As shown, RDIRL outperforms all other methods in terms of the mean cumulative reward, significantly outperforming the benchmark methods (i.e., AIRL, GCL, and GAIL).

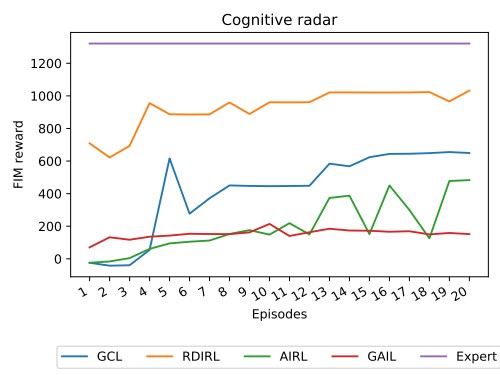

Figure 3: Learning curves for RDIRL and other methods.

## 7 CONCLUSIONS

We presented RDIRL within the IRL framework that generalizes recent advances in maximum entropy deep IRL to online settings. We first established the equivalence between upper bound loss function in equation 10 of the negative log likelihood in equation 2 to moment matching loss of (Swamy et al., 2021). Second, we leveraged sequential second-order Newton optimization to derive an online IRL algorithm by minimizing the moment matching loss function of equation 10 recursively and therefore established key theoretical properties of maximum entropy online deep IRL

RDIRL can learn rewards and cost functions online and greatly outperforms both prior imitation learning and IRL algorithms in terms of steps and samples required to converge. It generally reproduces the batch method's accuracy but in significantly less steps.

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

## A    APPENDIX

## B    EXPERIMENT DETAILS

In this section, we list down the implementation details of RDIRL and the baselines. The code is included in the supplementary material. We also report the hyperparameters used in the experiments, the detailed network architectures, training procedures and evaluation procedures used for our experiments.

## B.1 Training

In all our experiments, we use MPPI(Williams et al., 2016) as inner policy $q(\tau)$ in our baseline methods. MPPIis a probabilistic model predictive control policy that estimates an optimal action distribution that minimizes an agent's objective cost function. To do so, MPPI samples a number of trajectories and weighs these trajectories depending on how well they minimize the cost function, then updates the mean of its action distribution $q(\tau)$ accordingly. Since MPPI in an online policy , i.e it updates itself every time step, it makes it a natural choice of inner policy for online IRL problems, as we noticed in our preliminary experiments that it is much more stable and has faster convergence that traditional RL methods when implemented inside RDIRL.

The implementation of the baselines (GCL, AIRL,SQIL,ML-IRL and GAIL) are adapted from available public repository(HumanCompatibleAI, 2021).Furthermore, we adapt all the baselines to use MPPI as inner policy alongside our proposed approach. Since the inner policy is not SAC anymore like it was in the original baselines repositories, we tune the parameters of all the adapted baselines using grid search to produce best possible performance. The resulting parameters were used directly in RDIRL. We list the hyper-parameters of all the baselines used in different environments in Table 3. These hyper-parameters were selected via grid search.

Table 3: list of parameters used in each environment

| Environment | Learning rate | batch size | reward function updates | Nsteps | temperature | horizon | number of trajectories |
|---|---|---|---|---|---|---|---|
| Cartpole-v1 | $1e-4$ | 150 | 15 | 150 | 1e-3 | 50 | 2000 |
| MountainCar-v0 | $1e-4$ | 200 | 15 | 200 | 1e-2 | 85 | 3500 |
| HalfCheetah-v4 | $1e-4$ | 200 | 15 | 200 | 1e-2 | 50 | 500 |
| Walker2d | $1e-4$ | 200 | 15 | 200 | 1e-2 | 50 | 500 |
| Hopper | $1e-4$ | 200 | 15 | 200 | 1e-2 | 50 | 500 |
| Cognitive Radar | $1e-4$ | 200 | 10 | 200 | 1e-2 | 10 | 25 |

In all our experiments, we do multiple passes of parameter updates at the end of each episode using the Adam optimizer for all the baselines for best performance, except in our proposed approach RDIRL, since it is online. The number of passes is listed in the reward function update column of 3. The number of steps executed in each episode in listed in Nsteps column. Temperature,horizon and number of sampled trajectories are MPPI parameters.

PPO (Schulman et al., 2017) is used as the base MaxEnt RL algorithm for the expert policy. Adam is used as the optimizer.

In our proposed RDIRL, we use the same parameters of 3. Additionally, we use $P_{\theta_0} = 1e - 2I$ and $Q_\theta = 1e - 4I$ where $I$ is the identity matrix.

## B.2 Reward Function and Discriminator Network Architectures

We use the same neural network architecture to parameterize the cost-function/reward- function/discriminator for all methods. For continuous control task with raw state input, i.e. Cartpole,MountainCar, and the MuJoCo tasks, we use two-layer of MLP with ReLU activation function to pa- rameterized the cost function/discriminator with a hidden size of (16,16). Networks are randomly initialized at the start of each experiment, and all experiments are run on Nvidia-H200 GPU Cluster with 1 GPU per job(seed), with runtimes ranging from 30s/episode for CartPole and 2mins/episode for Walker2d on all benchmarked and competing IRL methods.

### B.3  RDIRL

RDIRL is recursive approach to deep inverse reinforcement learning (IRL), which incrementally estimates the parameters of a cost function from expert demonstrations. The method incorporates recursive updates inspired by Kalman filtering and quasi-Newton optimization, enabling efficient online learning from streaming data without requiring full-batch access to the dataset. The core algorithm is summarized in Algorithm 1.

The algorithm maintains a cost function $c_\theta(\tau)$ parameterized by $\theta$, which maps trajectories $\tau$ to scalar costs. The goal is to iteratively update $\theta$ such that trajectories generated from the current policy $q(\tau)$ match the expert demonstrations.

At each outer iteration (episode), we initialize the sampling policy $q(\tau)$ which can be a stochastic policy optimized with methods like PPO or MPPI, think of it as the IRL agent's best guess at mimicking the expert. Next, we initialize the parameter covariance $P_{\theta_0}$ along with a process noise term $Q_\theta$. $P_{\theta_0}$ represents the uncertainty over the parameters $\theta$ and $Q_\theta$ models uncertainty added to $\theta$ at each step (analogous to Kalman filtering).

The recursive nature of the algorithm is especially suited for online settings: instead of processing the entire expert dataset at once, RDIRL updates its internal model incrementally—one expert trajectory at a time. For each inner iteration, as soon as the algorithm observes one real expert demonstration $\tau_i^{\text{demo}}$, it samples a trajectory $\tau_i^{\text{samp}}$ drawn from $q(\tau)$.

We compute the gradients $\nabla_\theta c_\theta(\tau_i^{\text{demo}})$ and $\nabla_\theta c_\theta(\tau_i^{\text{samp}})$, which quantify how each trajectory influences the current cost estimate. Additionally, the algorithm computes (approximate) Hessians for both trajectories, which capture curvature information.

The parameter vector $\theta$ is then updated using a recursive rule:

$$\hat{\theta}(t_i) \leftarrow \hat{\theta}(t_{i-1}) - P_{\theta_i} \left( \nabla_\theta c_\theta(\tau_i^{\text{demo}}) - \nabla_\theta c_\theta(\tau_i^{\text{samp}}) \right),$$

where denotes the posterior covariance of the parameter estimate. This resembles a Kalman filter update, where the difference between expert and sampled gradients drives the parameter correction. $P_{\theta_i}$ is also recursively updated:

$$P_{\theta_i} \leftarrow \left[ (P_{\theta_{i-1}} + Q_\theta)^{-1} + \nabla_\theta^2 c_\theta(\tau_i^{\text{demo}}) - \nabla_\theta^2 c_\theta(\tau_i^{\text{samp}}) \right]^{-1}.$$

This equation accounts for new second-order information while controlling for process uncertainty.

After updating $\theta$, the sampling policy $q(\tau)$ is improved using any standard policy optimization method (e.g., PPO, MPPI), guided by the updated cost function $c_\theta$. This process continues over $K$ episodes, gradually aligning the agent's behavior with that of the expert.

### B.4  DERIVATION OF THE RECURSIVE SECOND-ORDER NEWTON SOLUTION

In a similar fashion to Kalman filtering optimization process described in (Humpherys et al., 2012), we seek to determine optimal solution $\Theta_N^* = \{\theta^*(t_0), \ldots, \theta^*(t_N)\}$ using the second-order Newton method sequentially, which recursively finds $\Theta_N^*$ given $\Theta_{N-1}^*$. To do so, we start by breaking the optimization function (11) as follows:

$$\mathcal{L}_i(\Theta_i) = \mathcal{L}_{i-1}(\Theta_{i-1}) + c_\theta(\tau_i^{\text{demo}}) - c_\theta(\tau_i^{\text{samp}}) + \frac{1}{2}\|\theta(t_i) - \theta(t_{i-1})\|_{Q_\theta^{-1}}^2. \tag{14}$$

Next, we further divide equation 14 into the following form

$$\mathcal{L}_i(\Theta_i) = \mathcal{L}_{i|i-1}(\Theta_i) + c_\theta(\tau_i^{\text{demo}}) - c_\theta(\tau_i^{\text{samp}}) \tag{15}$$

where

$$\mathcal{L}_{i|i-1}(\Theta_i) = \mathcal{L}_{i-1}(\Theta_{i-1}) + \frac{1}{2}\|\theta(t_i) - \theta(t_{i-1})\|_{Q_\theta^{-1}}^2. \tag{16}$$

Our optimization approach consists of minimizing equation 16 then minimizing equation 15 given equation 16 and the minimizer $\hat{\Theta}_{i|i-1}$ of equation 16. We proceed by minimizing equation 16 with respect to $\Theta_i$ by finding $\Theta_i$ that drives the gradient of equation 16 to zero. By taking the gradient of equation 16 with respect to $\Theta_i$ we obtain:

$$\nabla \mathcal{L}_{i|i-1}(\Theta_i) = \begin{bmatrix} \nabla \mathcal{L}_{i-1}(\Theta_i) - L_\theta^T Q_\theta^{-1}[\theta(t_i) - \theta(t_{i-1})] \\ Q_\theta^{-1}[\theta(t_i) - \theta(t_{i-1})] \end{bmatrix} \tag{17}$$

with $L_\theta = [0_{d_\theta \times d_\theta}, \ldots, 0_{d_\theta \times d_\theta}, I_{d_\theta \times d_\theta}]$ where $L_\theta \in \mathbb{R}^{d_\theta \times ((i-1) \times d_\theta)}$

Now, let the estimate $\hat{\Theta}_{i|i-1}$ of $\Theta_i$ be the minimizer of (16) obtained by setting $\nabla\mathcal{L}_{i|i-1}(\Theta_i)$ to zero, and note that $\hat{\Theta}_{i|i-1}$ can be broken as:

$$\hat{\Theta}_{i|i-1} = \begin{bmatrix} \hat{\Theta}_{i-1} \\ \hat{\theta}(t_{i-1}) \end{bmatrix} \tag{18}$$

Given equation 18 and equation 16, we proceed to minimize equation 15 using the second-order Newton update. We start by deriving the gradient of equation 15 as follows:

$$\nabla\mathcal{L}_i(\Theta_i) = \nabla\mathcal{L}_{i|i-1}(\hat{\Theta}_{i|i-1}) + \frac{\partial c_\theta(\tau_i^{\mathrm{demo}})}{\partial\theta} - \frac{\partial c_\theta(\tau_i^{\mathrm{samp}})}{\partial\theta}$$
$$= \begin{bmatrix} \nabla\mathcal{L}_{i|i-1}(\hat{\Theta}_{i|i-1}) \\ \frac{\partial c_\theta(\tau_i^{\mathrm{demo}})}{\partial\theta} - \frac{\partial c_\theta(\tau_i^{\mathrm{samp}})}{\partial\theta} \end{bmatrix} \tag{19}$$

For the sake of simplicity, let's define the following variables:

$$C_{\tau_{\mathrm{demo}}}^2(t_i) = \frac{\partial^2 c_\theta(\tau_i^{\mathrm{demo}})}{\partial^2\hat{\theta}(t_{i-1})}, C_{\tau_{\mathrm{samp}}}^2(t_i) = \frac{\partial^2 c_\theta(\tau_i^{\mathrm{samp}})}{\partial^2\hat{\theta}(t_{i-1})}$$

$$C_{\tau_{\mathrm{demo}}}(t_i) = \frac{\partial c_\theta(\tau_i^{\mathrm{demo}})}{\partial\hat{\theta}(t_{i-1})}, C_{\tau_{\mathrm{samp}}}(t_i) = \frac{\partial c_\theta(\tau_i^{\mathrm{samp}})}{\partial\hat{\theta}(t_{i-1})}$$

Therefore, at $\Theta_i = \hat{\Theta}_{i|i-1}$, equation 19 becomes:

$$\nabla\mathcal{L}_i(\Theta_i) = \begin{bmatrix} 0 \\ C_{\tau_{\mathrm{demo}}}(t_i) - C_{\tau_{\mathrm{samp}}}(t_i) \end{bmatrix} \tag{20}$$

Similarly, the Hessian of (15) is given by:

$$\nabla^2\mathcal{L}_i(\Theta_i) = \begin{bmatrix} \nabla^2\mathcal{L}_{i-1}(\Theta_{i-1}) + Q_\theta^{-1} & -L_\theta^T Q_\theta^{-1} \\ -Q_\theta^{-1} L_\theta & Q_\theta^{-1} + C_{\tau_{\mathrm{demo}}}^2(t_i) - C_{\tau_{\mathrm{samp}}}^2(t_i) \end{bmatrix} \tag{21}$$

Using the Newton second-order method, we can update our estimate of $\Theta_i$ given $\hat{\Theta}_{i|i-1}$ as follows:

$$\hat{\Theta}_i = \hat{\Theta}_{i|i-1} - \left(\nabla^2\mathcal{L}_i(\hat{\Theta}_{i|i-1})\right)^{-1}\nabla\mathcal{L}_i(\hat{\Theta}_{i|i-1}) \tag{22}$$

The resulting optimal variable $\hat{\theta}(t_i) \in \hat{\Theta}_i$ is given by equation 12. The procedure is repeated until $t_i = t_N$.

## B.5 ADDITIONAL EXPERIMENTS RESULTS

### B.5.1 ONLINE ADAPTATION OF COMPETING METHODS

In this section, we compare our proposed approach, RDIRL, with online-adapted versions of GAIL, AIRL,ML-IRL, and GCL. The online adaptation involves training each competing method using one expert demonstration at a time. Specifically, the loss function of each method is computed using a single observed expert sample at each time step, followed by an immediate update of the reward function neural network parameters. This process is repeated across the full episode of Nsteps.

As illustrated in Figure 4, our proposed method consistently outperforms the online-adapted baselines. Furthermore, the online adaptation does not significantly improve the performance of the original methods. In the case of Cartpole, it even leads to notable performance degradation and increased instability compared to both the original baselines (GAIL, AIRL,ML-IRL, GCL) and our approach, as shown in Table 4. These results highlight the advantage of our recursive optimization framework in producing more stable and accurate reward functions over naive online adaptation.

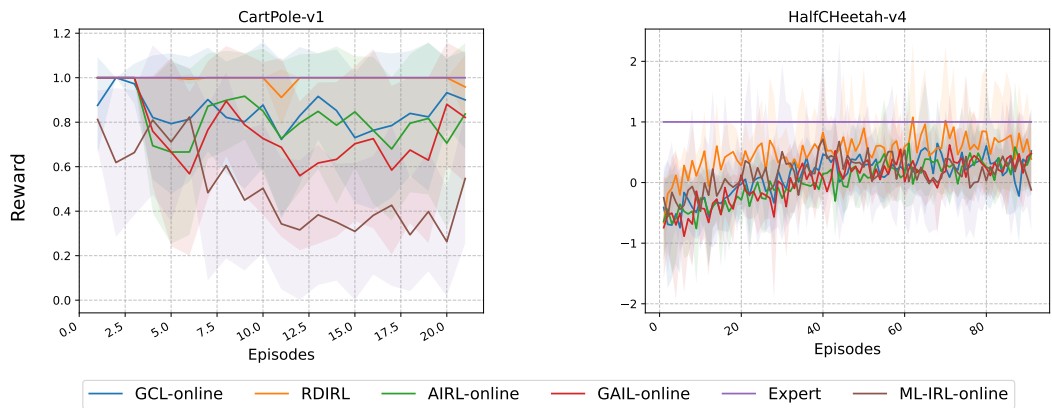

Figure 4: Learning curves for RDIRL and online adaptation methods.

Table 4: Comparison of mean reward values for different Gym environments and online adapted methods.

| Methods | CartPole | HalfCheetah-v4 |
|---|---|---|
| GAIL | $0.934 \pm 0.058$ | $-0.521 \pm 1.15$ |
| GCL | $0.92 \pm 0.09$ | $-0.226 \pm 1.27$ |
| AIRL | $0.953 \pm 0.069$ | $-0.54 \pm 1.11$ |
| GAIL-Online | $0.74 \pm 0.29$ | $0.02 \pm 0.51$ |
| GCL-Online | $0.84 \pm 0.25$ | $0.1 \pm 0.53$ |
| AIRL-Online | $0.81 \pm 0.26$ | $0.01 \pm 0.49$ |
| ML-IRL-Online | $0.49 \pm 0.29$ | $0.14 \pm 0.75$ |
| RDIRL (ours) | $\mathbf{0.99 \pm 0.13}$ | $\mathbf{0.49 \pm 0.59}$ |

