# OpenReview forum: "Recursive Deep Inverse Reinforcement Learning"
_ICLR.cc/2026/Conference — Submitted to ICLR 2026_

### Official Review · Reviewer_UCoK · 2025-10-31

**Soundness:** 2
**Presentation:** 2
**Contribution:** 3
**Rating:** 4
**Confidence:** 2

**Summary:**

The paper considers the problem of inferring the parameters of the cost function of an agent through observing the actions taken and the trajectories generated by the agent, referred to as inverse RL in the ML literature. The paper draws a parallel between estimating the parameters and the estimation problem addressed by the extended Kalman filter, and uses this for formulating a recursive algorithm using an upper bound. Numerical results on benchmark environments are used to illustrate the method.

**Strengths:**

The paper draws a parallel between the EKF used for state estimation in NLTI systems and the problem of IRL, which could be of interest. The focus on the online setting could make the paper interesting, although the evaluation does not focus on the online nature of the algorithm. Overall, the presented results indicate that the proposed scheme outperforms most baselines.

**Weaknesses:**

While the motivation for the paper appears to be adversarial settings and multiagent systems, the problem considered in the paper is limited to a single agent providing expert demonstrations, i.e., strategic behavior and coupling inherent to multiagent systems are not considered.

The considered setup assumes that for the purpose of IRL it is possible to interact with the environment using an arbitrary policy (in this case the inner policy q(\tau)).

The main idea of the paper is to treat online parameter estimation similar to the problem of state estimation in non-linear systems. At the same time, in the considered IRL setup the observations are sequences of actions and states (complete trajectories), which makes that there seems to be a conceptual mismatch between the EKF analogy and the considered IRL settting. Arguably, the problem is closer to system identification than to state estimation of a NLTI system, as the parameters of the cost function do not change. The main benefit of the recursive approach could be that past trajectories need not be maintained. This benefit is likely lost by the requirement of computing the Hessian for the recursive step. One could just as well treat the problem as a recursive least squares problem, the resulting algorithm would arguably be the same.

The presentation of the paper could be improved significantly. The p(\tau) is first referred to as a distribution, followed by entropy cost distribution, and finally as entropy. It is unclear what would be the interval [a,b] for y_n in (7). What do authors mean by replacing an equation by another equation, may be substitution? The impact of the choice N=M is not discussed in the derivation of (7), and it is unclear in what sense the proposed method would perform online IRL, considering that it operators on sequences of observations.

The numerical results compare the considered approach to a number of baselines. The most recent baseline seems to be from 2022. It would be good to compare to more recent work, e.g., task alignment or Bayesian IRL.

**Questions:**

How would the methodology be affected in the case of a strategic adversary that tries to avoid the cost function to be learned, or in the case of coupling in a multi-agent system?

What analytical guarantees can be provided, e.g., in an MDP setting, for the considered approach. How would these compare to those for ML IRL?

What is the benefit of treating IRL as a state estimation problem as opposed to a parameter estimation problem (solved using, e.g., recursive least squares)?

How does the proposed approach compare to more recent baselines?

---

### Official Review · Reviewer_1qwo · 2025-11-01

**Soundness:** 3
**Presentation:** 3
**Contribution:** 3
**Rating:** 8
**Confidence:** 2

**Summary:**

This paper proposes Recursive Deep Inverse Reinforcement Learning (RDIRL), an online method for inferring cost functions from observed behavior. It addresses the key limitation of standard deep IRL methods, which are typically offline and require large batches, making them unsuitable for real-time scenarios. The authors derive that a moment-matching loss function provides an upper bound for the standard maximum entropy (MaxEnt) IRL objective. This formulation enables the use of sequential, EKF-like second-order updates to process demonstrations incrementally. Experiments on benchmarks and an adversarial radar task show RDIRL converges significantly faster than baselines like GAIL, GCL, and AIRL.

**Strengths:**

The inspiration of this paper comes from the lack of high efficiency max-ent IRL algorithms, which is clear and natural. This paper provides a novel perspective that a specific moment-matching loss is an upper bound of the standard negative log-likelihood of max-ent IRL. The derivation connecting a moment-matching loss to an upper bound of the MaxEnt IRL objective is a key insight that enables the recursive algorithm. It seems that the proposed algorithm requires fewer samples compared with baselines. Both theoretical and empirical results demonstrate the advantage of proposed RDIRL.

**Weaknesses:**

The paper provides no analysis of the computational overhead of its second-order update (e.g., computing and inverting Hessians).

All competing methods (e.g., GAIL) were modified to use MPPI as their inner policy, rather than their originally designed policies (like PPO/SAC). This confounds the results, making it unclear if RDIRL is a better algorithm or just a better fit for MPPI.

More empirical results can enhance this paper, e.g. parameter sensitivity tests.

**Questions:**

Baseline Fairness: To validate the claims, could you provide a comparison where RDIRL (using MPPI) is benchmarked against the baseline methods using their original, intended inner policies (e.g., GAIL with PPO)?

---

### Official Review · Reviewer_BaSF · 2025-11-01

**Soundness:** 3
**Presentation:** 2
**Contribution:** 2
**Rating:** 4
**Confidence:** 2

**Summary:**

TLDR: This paper proposes a recursive Kalman filter like algorithm for IRL. The method integrates recursive second order updates into the learning phase of the cost function. The paper shows that this update can be done efficiently similar to a Kalman filter.

**Strengths:**

The main contribution of this work, in my potentially erroneous view is the recursive update. To the best of my knowledge, it is novel and I imagine would be of interest to the community. The experiments show great increase in performance in the particular setting the authors are interested (small network, few trajectories) in. This include standard Gym continuous control environment and an adversarial (am I correct in understanding it as a nonstationary environment?). The paper also include a small analysis of the results with smaller batch.

**Weaknesses:**

The upper bound in Section 4 is trivially true, with the upper bound $\mathcal{L}_{UB-MM}$ being actually the likelihood for exponential distributions (which include maximum entropy RL) (see section 3 of [1]).
The proposed method is a second order method and does not seem scalable. Is my understanding that the hessian of the reward model must be evaluated to update the preconditioning $P$ correct?

The experiments do not make it easier to disentangle the different parts of the algorithm. For instance, all experiments use MPPI instead of PPO which is not commonly used. Similarly, the network is very small (2 * 16 hidden layers, if I understand the notation correctly). This may be a consequence of the second order nature of this work or the motivating setting of this work. Regardless, I do not think the performance of RDIRL in these more "standard" settings matter but would help contextualize when the different component matter. From reading the paper, I can only say that RDIRL is a more suitable choice when
- I can tune the hyperparameters (no sensitive analysis has been done),
- The network must be tiny (which seems a senario where normal DRL may be hard to use),
- And, cannot use PPO.
Yet none of these are limitations of RDIRL but of the experiments.

[1] Wainwright, Martin J., and Michael I. Jordan. "Graphical models, exponential families, and variational inference." Foundations and Trends® in Machine Learning 1.1–2 (2008): 1-305.

**Questions:**

Would it be possible to fix the following non-score affecting typographical errors?
272 parenthesis missing around citation.
650 space after MPPI
651 space after MPPI
659 space around parenthesis
698 space around ",MountainCar,"
Please use superscript, $1e-4$ is not the same as $1e^{-4}$

Would it be feasible to extend the experiments by including the DeepMind control (DMC) environments as well?

At line 700, the runtime for a trajectory is reported somewhere around 30s to 2m. Is it possible to elaborate on what is being reported? Does this include parameter update?

Equation 19 isn't the gradient of L_(i|i-1)(hat(Theta)_(i|i-1) the same shape as the gradient of L_i(Theta_i)? If so should 19 be addition no concatenation? It seems that as this theorem is written, a different theta(t_i) is used to evaluate the cost and the the regularizer. Am I grossly misunderstanding the theorem?

How were the hyper-parameter chosen? Would it be possible to include sensitivity analysis?

How is figure 2 supposed to be interpreted? Is this the ratio of the estimated reward (or normalized reward?) compared to the actual reward?

Is it possible to add IQL?

---

### Official Review · Reviewer_TMEF · 2025-11-02

**Soundness:** 2
**Presentation:** 2
**Contribution:** 3
**Rating:** 2
**Confidence:** 4

**Summary:**

This paper introduces *Recursive Deep Inverse Reinforcement Learning* (RDIRL), an online deep IRL framework for inferring an agent’s or adversary’s underlying cost function in real time. The method upper-bounds the standard Maximum Entropy IRL objective with a tractable moment-matching loss that can be optimized recursively. Building on this reformulation, the authors develop a second-order, EKF-style recursive update for the cost network, allowing per-sample online learning without batch accumulation. With MPPI serving as the inner policy optimizer, RDIRL aims to achieve faster convergence and better sample efficiency. Experiments on continuous-control (CartPole, MountainCar, HalfCheetah, Hopper, Walker2d) and a cognitive radar task suggest improved performance over existing IRL and imitation-learning baselines.

**Strengths:**

* Reformulate the intractable MaxEnt IRL negative log-likelihood into a moment-matching loss where the expert trajectories are encouraged to be “cheaper” than sampled ones. This upper-bound form is additive over samples and naturally compatible with recursive EKF/Newton-style updates, making the method conceptually clear and practically appealing for online learning.

* Presents explicit recursive second-order update rules; the alternating updates between the outer IRL recursion and the inner MPPI control are well structured and directly implementable, making RDIRL an engineering-feasible framework for real-time inverse reinforcement learning. MPPI (Model Predictive Path Integral control) acts as a sampling-based short-horizon controller: at each time step it perturbs an action sequence, simulates rollouts, weighs trajectories by their cumulative costs under a temperature parameter, and updates the mean action accordingly. Because it requires no gradients and supports per-step replanning, it naturally fits the proposed online IRL setup.

**Weaknesses:**

* **Insufficient experimental coverage and missing classic baselines.**
  The experimental scope is narrow: only three MuJoCo environments (HalfCheetah-v4, Hopper, Walker2d) are tested, omitting widely used and more challenging benchmarks such as Ant, Humanoid, and Swimmer. In classic control, only CartPole and MountainCar are included.
  Moreover, RDIRL only compares against GCL, AIRL, GAIL, SQIL, and ML-IRL. It is recommended to include more methods like the classic Behavior Cloning (BC) method and the MaxEnt IRL methods such as Receding-Horizon IRL (Xu et al., 2022)，which mentioned in the paper.

* **Unfair training budget and metric convention.**
  All learning curves use “episodes” on the x-axis (Figures 1–2), with each episode containing very few environment steps (Table~3: CartPole=150; MuJoCo=200). This deviates from the community standard of aligning budgets by \textbf{environment steps}. Many IRL/IL baselines (especially RL-based ones) require longer rollouts for stable convergence; using “short episodes × few steps” systematically penalizes them. Training and evaluation budgets should be matched by environment steps, and curves plotted accordingly.

* The paper reports only normalized returns (Figures 1–2, Table 1) without providing the absolute expert reward values.

**Questions:**

* The core theoretical contribution (Eq. (6)–(10)) upper-bounds the MaxEnt-IRL NLL via a Matković–Pečarić-type inequality, assuming
  $$
  y_n = \frac{\exp(-c_\theta(\tau))}{q(\tau)} \in [a,b].
  $$
  However, the paper does not explain how boundedness is ensured or how $q(\tau)$ and $c_\theta$ are constrained or truncated. The text simply states “let $y_n$ be defined over $[a,b]$” (p.5), leaving the key condition unverified. When $q(\tau)$ is small or $c_\theta$ negative, $y_n$ can easily explode, invalidating or loosening the bound.
  Furthermore, the constant term $C = \log q(\tau) + K$ is treated as independent of $\theta$ (Eq. (9)), yet Algorithm 1 (step 11) updates $q$ each iteration, breaking the assumption of a fixed $q$ for the bound. Hence, the “fixed-policy upper bound” and the full training process may be inconsistent, undermining claims of monotonic improvement or unified objective optimization.

* In Table 1, ML-IRL shows negative normalized return on HalfCheetah and performs poorly on Hopper, whereas the NeurIPS 2022 ML-IRL paper reports near-expert performance on similar MuJoCo tasks (e.g., HalfCheetah  4472.85 vs. Expert 5098.30; Hopper  3121.68 vs. 3592.63).  Could the authors clarify the reason for this discrepancy? Without additional ablations or explanations, the current performance comparison remains difficult to interpret.

---

### Meta-Review · Area_Chair_mrbp · 2026-01-02

**Summary:**

Reviewers recognized the novelty of the recursive formulation of the maximum entropy inverse RL and a novel perspective on the moment matching formulation, which is of potential interest to the community. However, the reviewers also pointed out several limitations of the work (limited empirical validation, concerns about experimental setup, concerns about scalability of the proposed method) and asked reasonable clarification questions, which authors did not address during the rebuttal. For that reason, I recommend rejecting the paper.

I hope the detailed feedback provided by the reviewers would be helpful to the authors in revising and improving their manuscript before publishing it elsewhere.

**Reviewer Concerns:**

- Insufficient experimental validation, both in terms of environments used for evaluation and the baselines compared against.
- Concerns about fairness of the empirical comparison of the method against the baselines:
  - baselines were modified to use MPPI as the inner policy instead of the original policies the were designed for (PPO/SAC).
  - discrepancy between reported results for some of the baselines (ML-IRL) vs. results reported in the original NeurIPS paper.
- Lack of clarity on: (i) assumptions necessary for the theory to work, (ii) writing of different parts of the paper, (iii) scalability of the method.

**Reviewer Scores:**

The authors did not engage in the rebuttal, so I assume all the scores would've stayed the same.

---

### Decision · Program_Chairs · 2026-01-26

Reject